# A Genome-Wide Analysis of Nuclear Mitochondrial DNA Sequences (NUMTs) in Chrysomelidae Species (Coleoptera)

**DOI:** 10.3390/insects16020150

**Published:** 2025-02-02

**Authors:** Yeyan He, Siqin Ge, Hongbin Liang

**Affiliations:** 1College of Life Sciences, Shihezi University, Shihezi 832003, China; yeyanhe1997@163.com; 2Key Laboratory of Zoological Systematics and Evolution, Institute of Zoology, Chinese Academy of Sciences, Beijing 100101, China

**Keywords:** NUMT, Chrysomelidae, molecular markers, PCGs, phylogeny, gene flow, mitochondrial lineages

## Abstract

Nuclear mitochondrial DNA sequences (NUMTs) are mitochondrial DNA segments transferred to the nuclear genome, found in various insect groups. However, the unclear distribution of NUMTs in Chrysomelidae species complicates the selection of suitable molecular markers for species identification and phylogenetic analysis. This study investigates NUMT distribution across 32 Chrysomelidae beetle species, focusing on identifying mitochondrial protein-coding genes (PCGs) minimally affected by NUMTs and exploring their origin and evolutionary history using both PCG and NUMT data. Our results show that NUMTs are species-specific and chromosomally specific, with variable distribution patterns. *ATP8*, *COX1*, *ND1*, and *ND4* genes exhibit minimal NUMT interference. While phylogenetic conflicts arise from factors such as NUMTs, our findings suggest that NUMTs offer valuable insights into evolutionary history. Most NUMTs originate from distinct mitochondrial lineages, suggesting past mitochondrial diversification and gene flow. This study identifies molecular markers with minimal NUMT interference, providing useful tools for species identification, phylogenetic analysis, and diversity studies. Furthermore, our exploration of NUMTs highlights the dynamic integration of exogenous DNA into the nuclear genome, offering new insights into DNA transfer mechanisms and species evolution.

## 1. Introduction

Insect taxonomy encompasses a wide range of topics, including species identification, phylogenetic relationships, and biodiversity studies. The unique features of mitochondrial DNA (mtDNA)—maternal inheritance, region-specific evolutionary rate, and high copy number—led Hebert et al. to propose the *COX1* gene as a barcode marker for species identification [1]. Since then, mitochondrial genes have been widely used in phylogenetic reconstruction and biodiversity studies. The orthology of DNA sequences is the premise of establishing species barcodes or reconstructing phylogeny [2], which ensures data consistency and comparability. Mitochondrial genes, particularly the *COX1* gene, are effective for species identification and phylogenetic analysis, but their application is complicated by interference from nuclear mitochondrial DNA segments (NUMTs). The DNA barcoding project has faced criticisms [3], particularly regarding the co-amplification of NUMTs with universal [2] and sometimes specific primers [4]. NUMTs are DNA fragments that originate from the mitochondrial genome and integrate into the nuclear genome, classified as a special type of duplicated pseudogenes [5]. In 2023, Hebert et al. acknowledged that the *COX1* gene, as a species identification marker, had led to misidentification due to interference from NUMTs [6]. They examined the distribution of NUMTs with varying fragment lengths of the *COX1* gene across more than 1000 insect species and assessed how these segments contribute to the overestimation of species diversity [6].

The existence of NUMTs has introduced complexities in other research areas, including misidentification in barcode identification due to sequence similarity [6], interference with sequence alignment and erroneous phylogenetic tree construction [7,8,9], incorrect estimation of allele frequencies in population genetic analyses [10,11], and misleading geographic patterns in phylogeographic reconstructions [9,11,12]. Despite these limitations, NUMTs play a significant role in advancing the understanding of species evolution within a phylogenetic framework owing to their distinctive characteristics. One of the most notable roles of NUMTs is serving as “molecular fossils” that calibrate the timing of speciation due to their slower evolutionary rate compared to the mitochondrial genome [8,13]. Additionally, NUMTs have been used to root phylogenies, helping to reveal the early evolutionary relationships among species [14]. NUMTs also contribute to revealing the diversity of ancestral mitochondrial genomes and gene flow between different mitochondrial lineages, thus offering new insights into the evolutionary history of species [13,15,16]. Moreover, NUMTs in the same state at the same position are considered highly suitable for genetic studies as homoplasy-free markers, with their consistent presence reflecting ancestral relationships rather than convergent evolution [8,14,16,17]. Beyond their role in phylogenetic analysis, NUMTs also serve as valuable tools for studying mutation rates. Specifically, they can be used to compare mutation rates across nuclear genomes, thereby providing insights into the processes of genomic variation [14].

The study of NUMTs in insects began in 1983 when mitochondrial rRNA genes were identified in the nuclear genome of *Locusta migratoria* (Orthoptera: Acrididae) [18]. NUMTs research in insects initially focused on the order Orthoptera, likely due to their relatively large nuclear genomes [19,20,21]. As research expanded, NUMTs were also documented in the nuclear genomes of Coleoptera. The first study on NUMTs in Coleoptera, published in 2005, focused on Australian tiger beetles, where a NUMT derived from mitochondrial rRNA genes was identified [22]. Phylogenetic analysis of orthologous NUMTs along with their corresponding mtDNA lineages revealed that all NUMTs were monophyletic [22]. Bertheau et al. identified NUMTs in *Ips typographus* (Curculionoidea), specifically in the *COX1* gene, that differ by only one to three base pairs from authentic mitochondrial haplotypes, which they referred to as cryptic NUMTs [12]. However, *Dendroctonus valens* in the same family exhibited more diverse NUMTs from the *COX1* gene, including one with a 110 bp insertion [10]. Koutroumpa et al. demonstrated that the ambiguities observed in the sequencing results of the *COX1* and *COX2* fragments of *Monochamus galloprovincialis* (Curculionoidea) were not due to mtDNA heteroplasmy but rather caused by the presence of NUMTs [23]. Interestingly, no NUMTs were found in its sister species, *Monochamus sutor* [23]. Meštrović et al., in their phylogenetic study of certain species of the genus *Tribolium* (Tenebrionoidea), unexpectedly discovered a shortened sequence [24]. The sequence resulted from several internal deletions and may represent a nuclear paralogue of part of the *COX1* gene [24]. Frequent NUMT insertions have also been observed in the closely related species *Tribolium castaneum* [25]. Jordal et al. also found homologous sequences of the *COX1* gene derived from the nuclear genome in certain species of the Scolytinae subfamily of Curculionidae [26]. Haran et al. analyzed the sequencing electropherograms of 115 *COX1* sequences from *Monochamus galloprovincialis* (Cerambycidae) and identified slightly divergent pseudogenes in 49 of these sequences [11]. Sahoo et al. identified 80 NUMTs in *Zygogramma bicolorata* (Chrysomelidae), with lengths ranging from 33 to 3197 bp, during the assembly and analysis of mitochondrial and nuclear genomes [27].

Most studies on NUMTs in insects have focused on individual or partial mitochondrial gene fragments, such as *COX1*, *COX2*, and 16S rRNA genes, while only a few have examined all protein-coding genes (PCGs) [28,29]. To address this gap, the present study aimed to examine the distribution of NUMTs in 32 Chrysomelidae species, focusing on identifying mitochondrial PCGs with lower NUMT insertion frequencies. These genes can serve as reliable molecular markers for future research. Phylogenetic analyses were also conducted using two types of datasets: one comprising mitochondrial PCGs and the other consisting of *COX1* sequences combined with NUMTs. Both datasets provided valuable insights into the evolutionary history of these species, each from a different perspective.

## 2. Materials and Methods

### 2.1. Nuclear Genome and Mitochondrial Genome Data

Genomic data for 37 Chrysomelidae species, all assembled to the chromosomal level, were downloaded from NCBI. The GenBank accession numbers for both the nuclear genomes and corresponding mitochondrial genomes (mitogenome) of each species are provided in Table 1. To minimize interference from NUMTs, priority was given to mitochondrial genomes from four sources: (1) those assembled from whole-genome RNA-seq data, (2) those subjected to mitochondrial enrichment, (3) those obtained from multiple sequence alignments of the 13 mitochondrial PCGs from the same species, with >97% sequence similarity considered reliable, and (4) those submitted to the Darwin Tree of Life Project. Mitochondrial genome data from the Darwin Tree of Life Project (https://www.darwintreeoflife.org/ (accessed on 22 January 2025)) were processed using the MitoHIFI pipeline to remove potential NUMTs. These data are also considered as reference sequences [30]. Species listed as numbers 7, 8, 13, 26, and 34 in the table were excluded due to the unavailability of raw sequencing data and mitochondrial genomes or questionable assembly quality. Finally, 32 species were used for subsequent NUMT analysis.

### 2.2. Annotation and Extraction of Mitochondrial PCGs

For mitochondrial genomes with existing annotations on NCBI, the 13 PCGs were directly retrieved from the available annotations. For genomes lacking annotations, two annotation pipelines, MitoHiFi v3.2.2 [30,31] (https://github.com/marcelauliano/MitoHiFi (accessed on 1 December 2024)) and MITOS2 [32] (https://usegalaxy.org/root?tool_id=toolshed.g2.bx.psu.edu%2Frepos%2Fiuc%2Fmitos2%2Fmitos2%2F2.1.3%20galaxy0 (accessed on 1 February 2025)), were used to generate and integrate the results. *Drosophila melanogaster*, a well-studied model organism in insect research, was used as a reference. Its mitochondrial genome and annotation file (GenBank accession number: NC_024511.2) were downloaded to validate the annotation of the 13 PCGs. The nucleotide sequences of 33 species were translated into protein sequences using Geneious R9.0.2 [33], and multiple sequence alignments of homologous proteins were performed using Geneious R9.0.2. The start and stop codons of each gene were carefully verified based on the alignment results and available protein sequence data from NCBI.

### 2.3. NUMT Identification and Classification Based on Mitochondrial PCGs

NUMTs derived from the 13 PCGs were identified using BLASTN searches (E value = 1 × 10^-4^) against the nuclear genomes of the 32 species [34], focusing exclusively on sequences mapped to chromosomes and excluding those located on contigs or scaffolds. The positional distribution of NUMTs across chromosomes was visualized using Circos v0.69.8 [35] or ggplot2 v3.4.3 [36]. To extract NUMTs from different chromosomes, the Seqtk tool was used (https://github.com/lh3/seqtk (accessed on 1 December 2024)). The longest 13 PCGs among the 32 species were selected, and 70% of their lengths were used as the filtering threshold for the corresponding NUMTs. Duplicate NUMTs were removed using the SeqKit tool [37]. The remaining NUMTs were classified into five categories according to the following criteria: (1) C1-NUMTs: The length is consistent with the corresponding PCGs, with correct start and stop codons; (2) C2-NUMTs: NUMTs show significant length differences compared to the corresponding PCGs, with a length difference of more than 2 amino acids, while maintaining the correct reading frame; (3) C3-NUMTs: NUMTs display in-frame stop codons within the correct reading frame of the associated PCGs; (4) C4-NUMTs: NUMTs are 1 or 2 amino acids shorter or longer than the corresponding PCGs, with correct start and stop codons; and (5) C5-NUMTs: NUMTs show slight length differences compared to the corresponding PCGs, typically differing by 1 to 2 amino acids, with either the start or stop codons missing.

### 2.4. Phylogenetic Tree Reconstruction

Our study includes two types of datasets: the mitochondrial PCG dataset and the *COX1* dataset, which includes NUMTs (the *COX1*+NUMT dataset). The mitochondrial PCGs were used to construct five datasets: (1) the amino acid sequences of the 13 PCGs (the 13PCGs-AA) dataset, (2) the 11PCGs-AA dataset, which excludes *ATP8* and *ND6*, (3) the nucleotide coding sequences (CDS) of the 13 PCGs (the 13PCGs-CDS dataset), (4) the 11PCGs-CDS dataset, and (5) the *COX1* dataset. *Drosophila melanogaster* was used as the outgroup for all phylogenetic trees constructed. Before performing phylogenetic analysis, the substitution saturation was tested using DAMBE v7.3.32 [38]. If the index of substitution saturation (Iss) is not less than the critical Iss value (Iss.c), the sequences have undergone severe substitution saturation and are unsuitable for phylogenetic reconstruction [39]. The DAMBE analysis results indicated that *ATP8* and *ND6* were saturated (Iss > Iss.c, *p* > 0.05), suggesting that these genes exhibit substitution saturation levels that render them very poor for phylogenetics, according to the software’s criteria [38].

PhyloSuite v1.2.3 [40] supports both the supermatrix concatenation method and the multispecies coalescent method for phylogenetic analysis, along with a suite of tree-building tools that facilitate these methods [41]. Each dataset was aligned using the MAFFT tool [42] in PhyloSuite, followed by trimming with trimAI in PhyloSuite to automatically remove spurious sequences or poorly aligned regions from multiple sequence alignments [43]. IQTree v2.2.0 [44] in PhyloSuite was used to construct maximum-likelihood (ML) phylogenies with 1000 bootstrap replicates and the best-fit substitution model selected by ModelFinder [45] in PhyloSuite. The concatenation method requires a supermatrix, which is constructed by concatenating multiple genes from each species in a specific order. The multispecies coalescent method infers the species tree using Weighted ASTRAL v1.10.2.3 [46] in PhyloSuite, based on optimal single-gene trees. Two amino acid sequence datasets (the 13PCGs-AA and 11PCGs-AA datasets) were subjected to two phylogenetic analysis methods (concatenation and coalescent methods), while two nucleotide sequence datasets (the 13PCGs-CDS and 11PCGs-CDS datasets) were analyzed using only the concatenation method. The primary purpose of designing the nucleotide sequence datasets was to verify whether sequence type or saturation affects the construction of phylogenetic relationships using the concatenation method. Additionally, the *COX1* and *COX1*+NUMT datasets were analyzed using a single-gene phylogenetic analysis.

Bayesian inference (BI) trees were also constructed for four datasets (13PCGs-AA, 11PCGs-AA, 13PCGs-CDS, and 11PCGs-CDS) using the supermatrix concatenation method, following the same sequence processing procedures as used for ML tree construction. The processed sequences, along with their partitioning schemes and best-fit models, were imported into the MrBayes module of PhyloSuite. The parameters in the “Markov Chain Monte Carlo (MCMC) Settings” section were adjusted based on the criteria for assessing the convergence of the MCMC algorithm and the computational resources available. For small datasets, the default parameters in PhyloSuite were generally sufficient, while for larger datasets, the parameters were modified to set nchains = 16 and nruns = 4. The “Show MrBayes Data Block” option in PhyloSuite was used to generate and save the data as a “.nex” file.

To reduce computational costs, the ML tree (without bootstrap values) was used as a starting tree and added into the “.nex” file as a “tree block”, following the recommendations in the MrBayes manual [47]. The analysis was run on a multi-threaded Linux version of MrBayes v3.2.7a [47]. Convergence was assessed based on three metrics: the average standard deviation of split frequencies (ASDSF) falling below 0.01, the potential scale reduction factor (PSRF) approaching 1, and the estimated sample size (ESS) exceeding 200. If any of these criteria were not met, the number of generations (ngen) was increased, and the analysis was resumed from the last checkpoint.

### 2.5. Nucleotide Substitution Patterns and Selection Pressure Analysis of the COX1+NUMT Dataset

The Bensasson method can be applied to first assess nucleotide substitution patterns between paired sequences and then further estimate the number of nuclear integration events in NUMTs [48]. If two NUMTs originate from the same nuclear integration event, the nucleotide differences between them should be neutral, resulting in roughly equal substitution rates at the three codon positions [49]. The Bensasson method consists of two steps: (1) aligning each NUMT with its corresponding mitochondrial PCGs, removing frame-shifting insertions and deletions (InDels), and generating a “virtual” coding sequence; (2) calculating the number of mutations accumulated at each codon position and performing a chi-square test (df = 2, *p* < 0.05) to assess whether the mutations are evenly distributed across the three codon positions. [48,49]. Specifically, the chi-square test tested the null hypothesis that nucleotide substitutions occur at equal rates across the three codon positions in two sequences from the same mitochondrial lineage. NUMTs in the nuclear genome evolve neutrally and at a slower rate, so they retain characteristics similar to the ancestral mitochondrial genes [48,49]. In other words, when comparing the codon positions of a NUMT with its ancestral mitochondrial gene, the mutation rates across the three codon positions should be uniform. Likewise, for two NUMTs from the same mitochondrial lineage, the mutations at the three codon positions should also be evenly distributed. However, if one compares a NUMT to contemporary mitochondrial genes, and finds that the mutation frequency at the third codon position is significantly higher than at the first or second positions, this may indicate that the NUMT does not originate from the contemporary mitochondrial lineage. Similarly, nucleotide substitutions across the three codon positions of NUMTs from different mitochondrial lineages within the same individual should also be unevenly distributed. It should be emphasized that the Bensasson method requires optimization, as the chi-square test only assesses whether the mutations are evenly distributed across the three codon positions. Therefore, an additional check is needed to verify whether the number of mutations at the third codon position is greater than at the first and second positions. When a third codon position bias is observed between two sequences, they are considered to have originated from independent nuclear integration events [48,49]. Moreover, Baldo et al. described the possible origins of NUMTs and demonstrated that NUMTs with different origins are subject to varying selective pressures (Appendix A) [49]. The strength and mode of natural selection acting on protein-coding genes are usefully measured by the ratio of non-synonymous to synonymous substitutions (d_N_/d_S_) [50]. The PAML program CODEML was used to calculate the d_N_/d_S_ ratio [50]. Since CODEML is designed for PCGs, in-frame stop codons in “virtual” coding alignments should be edited as gaps. The input for CODEML included the final aligned sequences from the “virtual” coding alignments and the phylogenetic tree topology of the *COX1*+NUMT dataset. The phylogenetic tree provides the evolutionary history and ancestral relationships among species and is required for our CODEML analysis [50].

## 3. Results

### 3.1. Length and Codon Usage at the Start and Termination Sites of Mitochondrial PCGs

Analysis of the lengths of mitochondrial PCGs showed that *COX3* demonstrated the most stable length (789 bp), followed by *ND1* (948–954 bp), *ND3* (348–354 bp), *ATP8* (153–159 bp), and *CYTB* (1137–1143 bp). The usage of start codons was also examined. *ND1* predominantly uses TTG as the start codon, while the start codons of the other 12 PCGs are mainly ATN. However, the start codon of the *COX1* gene in *Bruchidius siliquastri* is unusual, as it is ACC. The start codons of the *ND4* and *NAD4L* genes all begin with the standard ATG, while most *COX3* and *CYTB* genes also use the ATG start codon. Regarding the usage of stop codons, *ND2*, *COX1*, *ND6*, and *ATP8* all terminate with the standard TAA stop codon. Both standard stop codons, TAA and TAG, are used in the *ND1*, *NAD4L*, and *CYTB* genes. In addition to TAA and TAG, the stop codons in *ND3*, *ND4*, *ND5*, *COX2*, *COX3*, and *ATP6* also include T and TA as stop signals. Species-specific differences are still observed, despite gene lengths being generally consistent within species of the same genus. For instance, the two species of the genus *Phyllotreta* exhibited identical gene lengths across all 12 PCGs, except for a one-amino-acid difference in the *ND6* gene. Detailed codon usage patterns and length differences are provided in Appendix A.

### 3.2. Distribution Patterns of NUMTs

The analysis of mitochondrial genome insertion regions across 32 species revealed that the nuclear genomes of 13 species contained fragments nearly as large as the entire mitochondrial genome (coverage > 98%). However, *Phyllotreta striolata* represented an extreme case, with only 5.53% of its mitochondrial genome regions inserted into the nuclear genome, indicating significant variation in the extent of mitochondrial genome insertions among different species (Appendix A). Analysis of NUMTs from different gene sources across different species reveals considerable differences in their quantities. Specifically, *Altica lythri* had 803 NUMTs, whereas *P. striolata* contained only a single *COX1* NUMT (Figure 1a). An internal correlation analysis of the quantities of NUMTs from different sources revealed that, except for the *ATP8*-NUMTs, the quantities of the remaining NUMTs showed a positive correlation trend of co-variation (Figure 1b). As *A. lythri* exhibited the highest number of NUMTs among the studied species, its mitochondrial PCGs and nuclear genes were visualized in a Circos plot (Figure 2). The plot shows the specific chromosomal locations where NUMTs have been inserted. Additionally, the plot includes information on the GC content of the surrounding regions and the positions and quantities of nearby genes. These additional details are included because previous studies have suggested that NUMTs tend to insert in non-coding regions [51] with higher AT content [52], which may influence their genomic integration patterns. The correlation analysis between the quantities of NUMTs from various gene sources and the nuclear genome size did not show a strong correlation. The highest correlation coefficient came from the *ATP8*-NUMTs, but it was only 0.35489 (Appendix A).

To investigate the distribution patterns of NUMTs on autosomes and sex chromosomes, species with annotated sex chromosomes were analyzed. NUMTs from different gene sources were visualized on the chromosomes within the cell nucleus (Figure 3 and Appendix A). The distribution patterns revealed that NUMTs can be distributed over the whole chromosome. For example, NUMTs of *Galeruca laticollis* were densely distributed across internal chromosomal regions, with some observed near the ends of multiple chromosomes (Figure 3). In certain regions, NUMTs exhibit localized clustering, with some NUMTs displaying high sequence similarity or even complete identity. The comparison of NUMT distribution between sex chromosomes and autosomes, using the Wilcoxon signed-rank test with Benjamini–Hochberg correction, revealed species-specific differences: while 9 species showed no significant difference in NUMT counts between sex chromosomes and autosomes, 10 species exhibited significant differences, indicating that the distribution of NUMTs on the X chromosome does not follow a consistent pattern across species (Figure 3 and Appendix A, Appendix A). Among the seven species with Y chromosome assembled, five species showed an almost absence of NUMTs derived from mitochondrial PCGs in the Y chromosome. However, the Y chromosomes of *Diorhabda carinulata* and *Lochmaea crataegi* displayed distinct NUMT distribution patterns. In *D. carinulata*, multiple large insertions (up to 10,731 bp) led to the clustering of NUMTs, with 46 NUMTs compressed into three regions shown in Appendix A. The Y chromosome of *L. crataegi* harbored an unusually high number of NUMTs, with a total of 73. The number of NUMTs on the Y chromosome of this species far exceeded those observed in other species, with many showing 100% sequence similarity (Appendix A).

### 3.3. Filtering Effects of NUMTs

An initial screening with a 70% length threshold significantly reduced the total number of NUMTs across 32 species, decreasing from 4565 to 1800. *ATP8*-NUMTs were only reduced by 13.56%, indicating relatively stable lengths, whereas *COX1*-NUMTs showed the highest reduction, dropping from 708 to 126 (82.20%). According to the classification results for each type of NUMT, clear evidence of functional decay is observed in the majority of NUMTs. C3-NUMTs contain in-frame stop codons, whereas inDels are observed in the sequences of categories 2 to 5 NUMTs (C2-C5 NUMTs) and some C3-NUMTs. When all C2-C5 NUMTs were removed from the total NUMT haplotypes, 8 of the 32 species became completely free from NUMT interference. The removal of C2-C5 NUMTs reduced the number of NUMTs per PCG from 107.62 ± 23.13 to 45.85 ± 10.64, achieving a filtering efficiency of 56.39% ± 8.16% (Appendix A). For each PCG, 52.40% ± 6.50% of species have successfully eliminated the influence of their corresponding NUMTs. For example, *ATP8*-NUMTs were filtered out in 20 species, while *COX1*-NUMTs were removed in 17 species out of the 32 analyzed (Appendix A). The remaining NUMTs were classified as C1-NUMTs, which shared similar lengths and ORF characteristics with the associated PCGs. In certain cases, their similarity was extremely high, reaching up to 100%. The proportion of C1-NUMTs with a sequence identity exceeding 97% (including 100%) of their corresponding PCGs was calculated. The results were as follows: *ATP6* (38.71%), *ATP8* (66.67%), *COX1* (46.67%), *COX2* (50.88%), *COX3* (35.39%), *CYTB* (60.53%), *ND1* (68.57%), *ND2* (52.27%), *ND3* (75.51%), *ND4* (65.85%), *NAD4L* (48.98%), *ND5* (57.78%), and *ND6* (61.91%).

Figure 4 presents a box plot illustrating the number of C1-NUMTs associated with each PCG across 32 species (Appendix A). As *ATP8* had the fewest C1-NUMTs, the significant difference in C1-NUMTs abundance between *ATP8* and other PCGs across these species was highlighted, as determined by the paired Wilcoxon signed-rank test for pairwise comparisons between *ATP8* and each of the other PCGs. As shown in Figure 4, the number of C1-NUMTs derived from *ATP8* was the smallest, and their distribution differed significantly from that of *ATP6*, *COX2*, *COX3*, and *ND3* (*p* < 0.05, Wilcoxon signed-rank test followed by Benjamini–Hochberg correction). Based on the boxplot analysis and the results of the Wilcoxon signed-rank test (Figure 4), *ATP8*, *COX1*, *ND1*, and *ND4* are minimally affected by NUMTs after filtration, suggesting that these genes can serve as relatively reliable molecular markers.

### 3.4. Phylogenetic Analysis

The comparison of species trees constructed from two amino acid (13PCGs-AA and 11PCGs-AA) and two nucleotide (13PCGs-CDS and 11PCGs-CDS) datasets showed that species within the same subfamily consistently grouped together. However, both the relationships among species within each subfamily and the topologies of inter-subfamily relationships varied across datasets. This pattern was also observed in the two gene trees, which were based on the *COX1* and *COX1*+NUMT datasets, respectively. In all phylogenetic trees, Chrysomelinae and Galerucinae consistently formed a sister group. In the species tree (Figure 5 and Appendix A), the clade of Chrysomelinae and Galerucinae showed strong support values, whereas the support values were relatively lower in the gene trees. For instance, the posterior probabilities of this clade were 1.00 in all BI trees (Figure 5b, Appendix A). In contrast, the bootstrap values for the clade in the two gene trees reconstructed using ML methods were below 75 (Appendix A). Similarly, Criocerinae and Bruchinae also formed a sister group in most phylogenetic trees, except for the two gene trees and the BI tree based on the 11PCGs-CDS dataset. However, the support values for the Criocerinae and Bruchinae clades were generally lower than those for the Chrysomelinae and Galerucinae clades. The topologies derived from the same dataset, constructed using both ML and BI tree-building methods, were generally consistent. In particular, the 11PCGs-AA dataset yielded identical topologies across both methods (Figure 5). The ML trees for the 13PCGs-AA (Appendix A) and 11PCGs-AA datasets (Figure 5a), constructed using the supermatrix concatenation method, exhibited consistent topologies with only slight differences in their bootstrap support values. The ML trees constructed for these two datasets using the multispecies coalescent method showed the same patterns (Appendix A). The ML trees constructed from the two nucleotide sequence datasets showed relatively large variations in bootstrap values and slight differences in their topologies, specifically in the positions of *Neocrepidodera transversa* and *Hermaeophaga mercurialis* (Appendix A). The BI trees constructed from the 13PCGs-AA and 11PCGs-AA datasets differed topologically only in the placement of *Chrysomela aeneicollis* (Figure 5b; Appendix A). The BI trees based on the 13PCGs-CDS and 11PCGs-CDS datasets exhibited topological inconsistencies in the placement of *Crioceris asparagi* and in the internal relationships within the clade comprising *Lochmaea capreae*, *L. crataegi*, and *Galerucella nymphaeae*. (Appendix A). According to Kulkarni et al., phylogenetic relationships derived from nucleotide and amino acid data are often inconsistent [53], a pattern that is also reflected in our results. Furthermore, nucleotide and amino acid datasets show notable differences in support values at the same nodes. For example, large differences in bootstrap values were observed when comparing the same nodes in Figure 5a and Appendix A. Specifically, the node consists of Chrysomelinae, Galerucinae, Criocerinae, and Bruchinae (Δ bootstrap value = 44). Compared to the amino acid-based ML trees, the nucleotide-based ML trees exhibit higher support values at the leaf nodes and lower support values at nodes closer to the root. However, no clear trend was observed in the BI trees.

The phylogenetic relationships inferred from the *COX1*+NUMT dataset showed inconsistencies with those inferred from the *COX1* dataset, both in subfamily-level and species-level topology (Figure 6, Appendix A). For instance, in the ML tree based on the *COX1* dataset, Cassidinae and Criocerinae formed a clade. However, in the *COX1*+NUMT dataset, Criocerinae was the most basal subfamily, while Cassidinae and Cryptocephalinae formed a distinct clade. Additionally, in the *COX1*-based ML tree, Cryptocephalinae clustered with the Chrysomelinae and Galerucinae clades, with a bootstrap value of 78. In contrast, in the *COX1*+NUMT dataset, Cryptocephalinae, Cassidinae, Chrysomelinae, and Galerucinae formed a clade but with a much lower bootstrap value of 11. Moreover, inconsistencies in phylogenetic relationships were not only observed at the subfamily level but also the species level, such as among the three species of the genus *Chrysolina*. Furthermore, the variation in the phylogenetic relationships was also reflected in the considerable variation in bootstrap values within both gene trees, with the *COX1* dataset ranging from 19 to 100 and the *COX1*+NUMT dataset ranging from 11 to 100. These inconsistencies in subfamily-level and species-level relationships between the *COX1* and *COX1*+NUMT datasets provide insights into the potential impact of NUMTs on phylogenetic tree construction. To explore this further, we examined the topological structure and branch lengths of NUMTs in relation to their corresponding *COX1* genes. The phylogenetic tree based on the *COX1*+ NUMT dataset showed that the NUMTs of *Diorhabda carinata* and *D. carinulata* were split into two distinct clades. One clade included the *COX1* sequences of both species, along with some of their corresponding NUMTs, while the other clade contained the remaining NUMTs from both species. The NUMTs of the remaining species were grouped with their associated *COX1* sequences into a larger clade. Within the same species, multiple NUMTs often formed distinct subclades. Examination of the branch lengths in the ML tree revealed that the branch lengths of NUMTs are either shorter, equal to, or occasionally longer than those of the associated *COX1* sequences.

### 3.5. Selection Pressure of the COX1+NUMT Dataset

According to the Bensasson method, NUMTs were compared with their corresponding PCGs from the same species by analyzing mutations at the first, second, and third codon positions. The accumulated mutation counts at the three codon positions and the chi-square test results are recorded in Appendix A. Some NUMTs showed an uneven mutation distribution, displaying a third codon position bias similar to that observed in mitochondrial PCGs, while the other NUMTs did not exhibit this bias. These observations may suggest that two categories of NUMTs originate from different evolutionary backgrounds and are subject to distinct selective pressures. The following hypotheses were formulated: The null hypothesis (H_0_) assumed that both the *COX1* gene and NUMTs of the species were subject to the same selective pressures, while the alternative hypothesis (H_1_) posited that these sequences were influenced by three distinct selective pressures. The results of the CODEML analysis suggested that H_0_, which assumed all branches were subject to a single selective pressure, yielded a d_N_/d_S_ value of 0.01543 and a log-likelihood score of −8987.494213. In contrast, H_1_, which allowed for three distinct selective pressures on the branches, yielded a log-likelihood score of −8807.924057. A chi-square test comparing the two log-likelihoods (LnL1 − LnL0 = 359.140312, df = np1 − np0 = 2) resulted in the rejection of H_0_ and acceptance of H_1_ (*p* < 0.01). The results of H_1_ supported the presence of three distinct d_N_/d_S_ values: 0.00573 for mitochondrial functional gene branches, 0.04739 for blue branches, and 0.22415 for red branches. The d_N_/d_S_ value is commonly used to assess the intensity and type of selective pressure acting on gene sequences [54]. Hypothesis testing using CODEML supported H_1_, indicating that the *COX1* gene is primarily influenced by strong purifying selection from the mitochondria, as evidenced by the d_N_/d_S_ value significantly less than 1. In contrast, the highest d_N_/d_S_ value indicates that NUMTs, whether derived from the species’ *COX1* gene or amplified through nuclear replication, are primarily influenced by nuclear genomic selective pressures. NUMTs with d_N_/d_S_ values between the maximum and minimum reflect mixed selective pressures. The d_N_/d_S_ values of both categories of NUMTs are higher than those of mitochondrial functional genes, suggesting a release of functional constraints in NUMTs.

## 4. Discussion

### 4.1. Conservation and Diversity of Start and Stop Codons and Sequence Lengths in the 13 PCGs

PCGs in mitogenomes are comparatively conserved in start codons, stop codons, and sequence length. This conservatism is clearly observed in the genera *Diabrotica* and *Diorhabda*, with identical lengths and codon usage of all 13 PCGs across species within each genus. Mitochondrial PCGs of most species in the Chrysomelidae family studied here predominantly use the standard ATG start codon or other ATN variants. The termination codons are typically TAA or TAG, with some incomplete stop codons (T or TA) present. Incomplete stop codons (T or TA) are commonly found in the mitochondrial genomes of metazoans, and they are thought to be completed to TAA through post-transcriptional modification at the 3′ end [55]. The use of TTG as the start codon for *ND1* is not uncommon within Coleoptera, reported not only in Chrysomelidae but also in Tenebrionidae [56], Curculionidae [57], and Cerambycidae [58]. The most frequent start codon in the *COX1* gene is ATT, followed by ATC. The use of ACC as the start codon for the *COX1* gene in *B. siliquastri* is highly species-specific. Although ACC is relatively rare as a mitochondrial start codon within Coleoptera, it has been reported multiple times in other orders, particularly in Orthoptera [59,60,61,62,63]. The lengths of the five genes *COX3*, *ND1*, *ND3*, *ATP8*, and *CYTB* are relatively conserved across the 32 Chrysomelidae species, with a difference of no more than two amino acids between species for each gene. In contrast, the remaining eight genes show greater length diversity. For example, the two species of the genus *Phyllotreta* had the shortest *ND5* gene (1668 bp), while *P. cochleariae* had the longest *ND5* gene (1722 bp). Schneider et al. suggested that mitochondrial genes, compared to their prokaryotic ancestors, generally show a reduction in length, which is linked to the overall decrease in genome size, suggesting that smaller genomes may offer a selective advantage [64]. Therefore, the variation in *ND5* gene length between species, such as the difference between 1668 bp and 1722 bp, could be considered an adaptive change driven by evolutionary pressures. This change may be associated with the overall reduction in mitochondrial genome size and its functional optimization. To further distinguish these gene variations from NUMTs, researchers can more accurately infer and identify NUMTs by assessing the presence of characteristic open reading frames (ORFs) similar to those of true genes.

### 4.2. The Distribution of NUMTs and the Selection of Molecular Markers

Early studies often considered NUMTs to be abnormal and rare events [18], usually short in length [25,34]. However, an increasing body of research suggests that the occurrence of NUMTs is widespread [20], and the frequency of large NUMTs is also high [52,65]. Although many studies suggest a positive correlation between the number of NUMTs and nuclear genome size [8,20], no such trend was observed in the present study (Appendix A). Our results are consistent with those reported by Ding et al. for bumblebees [28]. It was unexpected that several nearly complete mitochondrial genomes were integrated into the nuclear genome (Appendix A). The analysis of NUMT insertion sites across different chromosomes within the same species reveals an uneven distribution, with NUMTs frequently clustering in specific genomic regions. Investigation into the similarity and origins of these clustered NUMTs suggests that their accumulation may result from localized chromosomal duplication events, leading to NUMT family expansions, or the insertion of large mitochondrial DNA fragments. Alternatively, this uneven distribution may be associated with particular chromosomal structures, such as pericentromeric heterochromatin, known for its low transcriptional activity, high repeat content, and structural instability [66]. Wang et al. observed a pattern of variation in AT content surrounding NUMTs, where the AT content increased progressively as the distance to NUMTs decreased [52]. Multiple large mitochondrial DNA insertions have also been identified on the human Y chromosome [67], a phenomenon similar to what was observed in the Y chromosome of *D. carinulata*. NUMTs are generally less abundant on the Y chromosome, which may be attributed to the shorter length and recombination suppression characteristics [68]. The Y chromosome of *D. carinulata* contains an *ND1*-NUMT identical to one found on an autosome, suggesting that recombination between the Y chromosome and an autosome may have occurred. However, this phenomenon is observed at a significantly lower frequency compared to autosomes. Despite significant variation in the distribution of NUMTs among different species (Figure 1, Figure 3, and Appendix A), a relatively consistent trend is observed across all species: *COX1*-NUMTs are the most frequent, while *ATP8*-NUMTs are the least frequent. Ožana et al. reported comparable statistical results in their study of NUMTs in three species of Odonata [29]. Likewise, Ding et al. quantified the number of NUMTs derived from various genetic sources in 17 species of the genus *Bombus* (Hymenoptera) and also found that *COX1*-derived NUMTs were the most common [28].

NUMTs were characterized based on the presence of ORF features corresponding to mitochondrial PCGs. C1-NUMTs have ORF features resembling those of their corresponding mitochondrial PCGs, making them more challenging to filter. In contrast, C2-C5 NUMTs are relatively easier to filter. The best filtering results are observed for *ATP8*, *COX1*, *ND1*, and *ND4*. The next group includes *CYTB*, *ND2*, *ND5*, and *ND6*, which show moderate filtering efficiency. The poorest filtering results are seen in *ATP6*, *COX2*, *COX3*, *NAD4L*, and *ND3*, where a higher proportion of NUMTs remained (Figure 4). A conserved length and a relatively low frequency of NUMTs were found in the *ATP8* gene among species in the Chrysomelidae family studied, indicating its potential for distinguishing closely related species. Bonin et al. explored the optimal sequence similarity thresholds for clustering molecular operational taxonomic units (MOTUs) in DNA metabarcoding studies, suggesting that a threshold of 96–99% is suitable for universal markers [69], with the 97% threshold being widely applied in insect DNA barcoding research [70,71]. Consistent with these findings, our analysis shows that the p-distance of the *ATP8* gene between species within the same genus is consistently greater than 3%. For example, the sequence identity of the *ATP8* gene between *Phyllotreta cruciferae* and *P. striolata* (71.95%), and among the three *Diorhabda* species, ranges from 72.84% to 96.82%. Lee et al. have suggested that the *ATP8* gene, with its lower intraspecific and higher interspecific genetic divergences, is a suitable molecular marker for species identification in the Eriosomatini (Eriosomatinae) and potentially other insect groups, improving the accuracy of DNA barcoding [72]. On the other hand, substitution saturation is easily reached in *ATP8* due to its short length and relatively high evolutionary rate, which limits the applicability of its nucleotide pattern for phylogenetic analysis. A similar issue is observed with the *ND6* gene. However, the use of incomplete gene sequences, especially *COX1* barcodes that exclude the start and stop codons of the *COX1* gene, reduces the ability to accurately identify NUMTs. Kaya et al. demonstrated that both the diversity and frequency of NUMTs decrease as sequence length increases [21]. The suggestion by Ožana et al. to use the *ND1* gene alone or in combination with *COX1* as molecular markers for Odonata was supported by their findings [29]. Similarly, Pereira et al. aligned whole-genome sequencing data to a reference mitochondrial genome and found that *CYTB* and *COX3* were regions of low diversity and coverage in *Chorthippus parallelus* (Orthoptera), advocating for their use as new molecular markers for this species [19].

### 4.3. Origin and Evolution of NUMTs

Our analysis showed four main phenomena concerning NUMTs: (1) some NUMTs have less than 97% identity with their associated mtDNA genes, (2) the ML tree, based on the *COX1*-NUMT dataset, displays unusual topological structures and branch length patterns, (3) codon position bias has been observed in most NUMTs, and (4) three distinct d_N_/d_S_ values are observed: the *COX1* gene exhibit the lowest d_N_/d_S_ values, NUMTs with a mixed evolutionary history show intermediate d_N_/d_S_ values, and NUMTs subject only to nuclear genome selection pressure have the highest d_N_/d_S_ values. These results are consistent with the findings of Baldo et al. [49]. The evolutionary rate of mitochondrial genomes in metazoans is roughly 10 times faster than that of the nuclear genome [16]. This difference may explain why some NUMTs have shorter branch lengths than the *COX1* gene. NUMTs with longer branch lengths, compared to the corresponding *COX1* gene, tend to have a third codon position bias. These NUMTs with median d_N_/d_S_ values, originating from unsampled or extinct mtDNA lineages outside the species studied, reflect a mixed evolutionary history involving both mitochondrial and nuclear genomes [49]. The recovery of multiple NUMTs of distinct origins from a single individual suggests that the nuclear genome contains ancient mtDNA fragments incorporated at different stages of the species’ evolutionary history [13]. A total of 66 independent nuclear integration events of the *COX1* gene were estimated to have occurred across 32 species. This estimate relied on the Bensasson method, which may overestimate the number of events, particularly when the total number of mutations is small (Appendix A). For example, the chi-square test rejected the hypothesis of uniformity in codon positions for the 18-*COX1* numt1 and 18-*COX1* numt7 sequences (1st = 1, 2nd = 2, 3rd = 7). Moreover, the high identity (over 99%) between these two sequences further suggests that they may originate from the same nuclear integration event rather than multiple independent events.

*COX1* sequences and their corresponding NUMTs generally cluster together on the phylogenetic tree, except for *D. carinata* and *D. carinulata* (Figure 6). NUMTs of more recent origin typically form sister groups with their corresponding *COX1* sequences within species, whereas older NUMTs tend to cluster at the base of the phylogenetic tree of the clade. Most NUMTs originate from distinct mitochondrial lineages. By contrast, the integration of the *COX1* gene from the species’ own mitogenome into the nuclear genome is rare among the species studied, occurring only in *Plagiodera versicolora*, *G. laticollis*, *G nymphaeae*, *Psylliodes chrysocephala*, and *L. decemlineata*. Many NUMTs, after being integrated into the nuclear genome, evolve through replication and mutation, forming multiple NUMT families [49]. This phenomenon, particularly pronounced in *A. lythri*, may be one of the factors contributing to the observed NUMT burst in this species. In particular cases, the distribution pattern of NUMTs may be influenced by gene flow events. The *COX1* genes of *D. carinata* and *D. carinulata* group with 5-*COX1* numt1 and 6-*COX1* numt1, while the remaining NUMTs from both species form a separate clade, constituting a distinct NUMT family. The main branch of this NUMT family clade is significantly longer than the *COX1* gene branches of *D. carinata* and *D. carinulata*, strongly suggesting that this NUMT family has a mixed evolutionary history. The presence of NUMTs at the same loci across multiple species may indicate phylogenetic relationships [16]. An additional check was conducted in this study, but the results did not support the hypothesis that this NUMT family originated from a nuclear integration event in the common ancestor of the two species. These two pieces of evidence together support the plausible conclusion that, during their evolutionary history, another mitochondrial lineage likely hybridized with both *D. carinata* and *D. carinulata*. Overall, these results highlight the complexity and dynamic nature of mitochondrial lineage diversity, providing new insights into the evolutionary history of the species.

### 4.4. Phylogenetic Conflicts and Evolution History

Differences between amino acid-based and nucleotide-based trees suggest that sequence type and saturation levels may impact phylogenetic signals, leading to biases in both topology and support values (Figure 5 and Appendix A). Translating nucleotide sequences into amino acid sequences is a common approach to reduce the effects of substitution saturation [73]. The failure to recover monophyly within genera and the instability in the positions of species can be attributed to two factors: (1) the reduction in phylogenetic signal in amino acid trees [74] and (2) ILS resulting from recent speciation events [75]. The hypothesis of ILS may be supported by two independent coalescent analyses for 13 PCGs-AA and 11 PCGs-AA datasets, with a final normalized quartet score of 0.82 and 0.84, respectively. Subfamily-level phylogenies based on *COX1*+NUMT and *COX1* alone show conflicting patterns (Figure 6, Appendix A). This discrepancy may be attributed to the significant differences in mutation rates between nuclear and mitochondrial genomes. When both genomic sources are treated as parallel markers in phylogenetic analysis, the complexity of their contrasting evolutionary rates complicates the interpretation of the phylogenetic tree [14]. Our phylogenetic analyses are primarily based on 13 mitochondrial PCGs, with some studies also utilizing 13 PCGs, and individual nuclear genes or morphological data [76,77,78,79]. While some results are consistent, they highlight the limitation of drawing stable conclusions from a small number of genes or morphological data alone. The limited number and narrow functional scope of mitochondrial PCGs fail to capture the full complexity of a species’ biological functions. Additionally, the phenomenon of NUMTs should also be considered when using mitochondrial genes for tree construction. In cases of prevalent gene tree incongruence, the multispecies coalescent method is better equipped to handle gene tree discordance caused by ILS compared to the supermatrix concatenation method [46].

## 5. Conclusions

The genome-wide distribution of NUMTs in Chrysomelidae species is chromosomally specific, species-specific, and widespread. A discernible distribution pattern is found, where the number of *COX1*-NUMTs is highest and *ATP8*-NUMTs is lowest in most species. The method of examining gene length and ORF features effectively filters out most NUMTs, particularly for the genes *ATP8*, *COX1*, *ND1*, and *ND4*. The *CYTB*, *ND2*, *ND5*, and *ND6* genes show moderate performance, while NUMTs in the other PCGs are the most difficult to filter out. The analysis of the relationship between *COX1* and its NUMTs suggests that most NUMTs originate from distinct mitochondrial lineages, reflecting the diversity of past mitochondrial lineages and gene flow in the species’ evolutionary history. Compared to other datasets, the *COX1*+NUMT dataset yields more inconsistencies in the phylogenetic relationships at the subfamily and species levels, with greater differences in topology and bootstrap values. The combined use of NUMTs and mitochondrial genes in phylogenetic tree construction assumes identical evolutionary rates for both, which can lead to inaccuracies in the inferred phylogenetic relationships. Our study also investigates the impact of sequence type, sequence saturation, and tree-building methods on the phylogenetic results based on the mitochondrial PCG dataset.

Understanding the presence of NUMTs in different taxa is essential, as it not only helps assess the potential risks in DNA barcoding but also provides a new perspective on the evolutionary history of species. Therefore, optimizing experimental procedures and strengthening quality control methods are necessary. It is advisable to prioritize the use of longer or more complete mitochondrial PCGs and avoid the use of *rRNA* [6]. The detection methods for NUMTs primarily involve the identification of PCR ghost bands, sequence ambiguities in chromatograms, in-frame stop codons or inDels, and extra bands in restriction enzyme profiles [14,21]. For specimens of ancient origin, it is advisable to incorporate a DNA dilution step in the PCR procedure [80]. This approach is both effective and cost-efficient. For fresh specimens, RNA can be extracted for RT-PCR or directly sequenced, or, alternatively, DNA can be used for mitochondrial genome assembly. If the experimental budget allows, third-generation sequencing is an excellent method to avoid the disassembly of NUMTs into the mitochondrial genome [30]. Most NUMTs are relatively rare and individual-specific [81]. Therefore, it is essential to increase the sampling effort of species from different geographical regions to reduce experimental error. Bioinformatics tools like MetaWorks v1.13.0 [82,83] and metaMATE v0.4.3 [84] are essential for NUMT filtering in large-scale projects. Beyond NUMT filtering, more comprehensive approaches are necessary for improving phylogenetic accuracy. In particular, the use of hundreds of functional genes and multispecies coalescent methods is emerging as a key trend for obtaining more accurate phylogenetic relationships [85].

## Figures and Tables

**Figure 1 insects-16-00150-f001:**
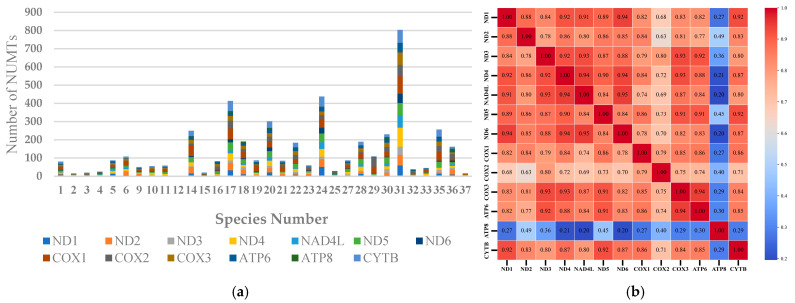
Distribution patterns of nuclear mitochondrial DNA sequences (NUMTs) in Chrysomelidae species. (**a**) Types and numbers of NUMTs. The specific species represented by the species number can be found in Table 1; (**b**) correlation analysis of NUMTs from various gene sources.

**Figure 2 insects-16-00150-f002:**
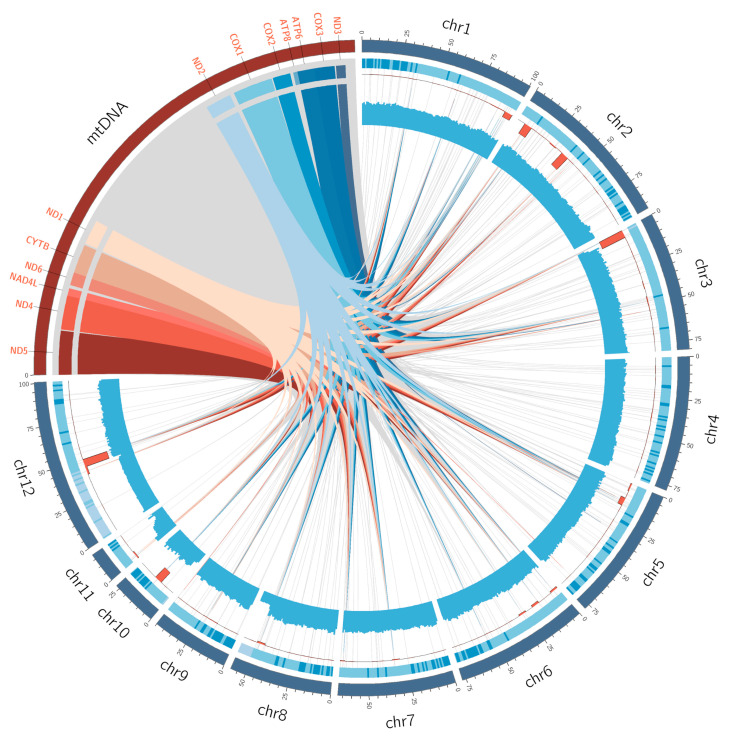
Mitochondrial genome insertion into the nuclear genome in *Altica lythri*. The outermost circle represents the genome, with red indicating the mitochondrial DNA (mtDNA) and blue indicating the nuclear genome. The lines within the mtDNA, varying in shades of red and blue, represent the insertion sites of 13 protein-coding genes (PCGs) into the nuclear genome. Gray regions represent mtDNA fragments that have undergone insertion events, and gray lines indicate the specific nuclear genome locations where these mtDNA fragments were inserted. In the nuclear genome, the second circle (light blue heatmap) represents gene density, showing the number of genes per 1 Mb; the darker the color, the higher the gene density. The third circle (orange histogram) shows the density of NUMTs, indicating the number of NUMTs per 1 Mb. The fourth circle (blue histogram) represents GC content, displaying the GC percentage per 1 Mb.

**Figure 3 insects-16-00150-f003:**
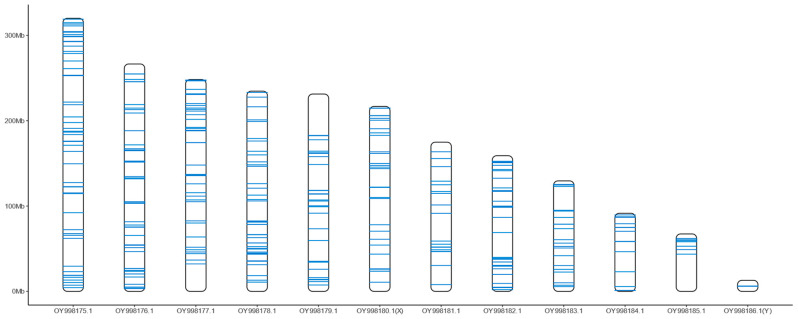
Chromosomal localization of NUMTs derived from 13 mitochondrial PCGs in *Galeruca laticollis*. Blue horizontal lines represent the insertion sites of NUMTs. Due to the compression of chromosome lengths along the *y*-axis, closely located NUMTs may appear as a single blue line.

**Figure 4 insects-16-00150-f004:**
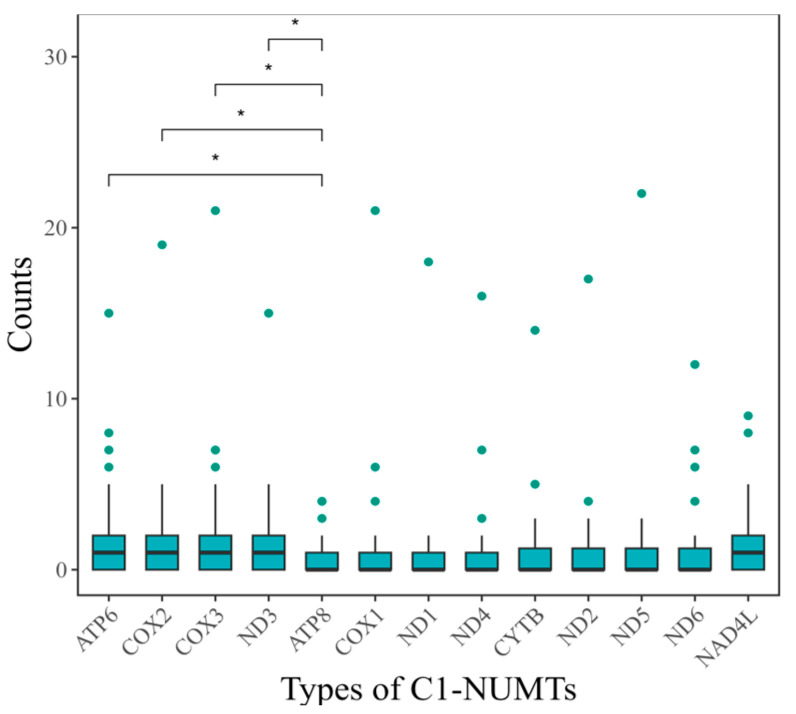
Distribution and significance of category 1 NUMTs (C1-NUMTs) from *ATP8* compared to each of the other mitochondrial protein-coding genes (PCGs) in Chrysomelidae species. The asterisk (*) denotes a statistically significant result with *p* < 0.05, as determined by the paired Wilcoxon signed-rank test with Benjamini–Hochberg correction.

**Figure 5 insects-16-00150-f005:**
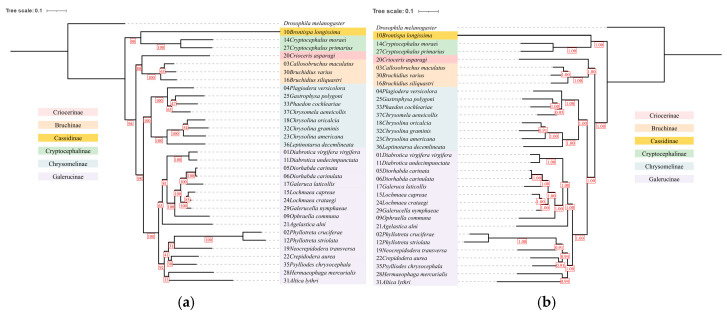
Phylogenetic trees based on the 11PCGs-AA dataset using the supermatrix concatenation method. (**a**) Maximum Likelihood (ML) tree, with red-boxed numbers indicating node support as bootstrap values; (**b**) Bayesian inference (BI) tree, with red-boxed numbers indicating node support as posterior probabilities.

**Figure 6 insects-16-00150-f006:**
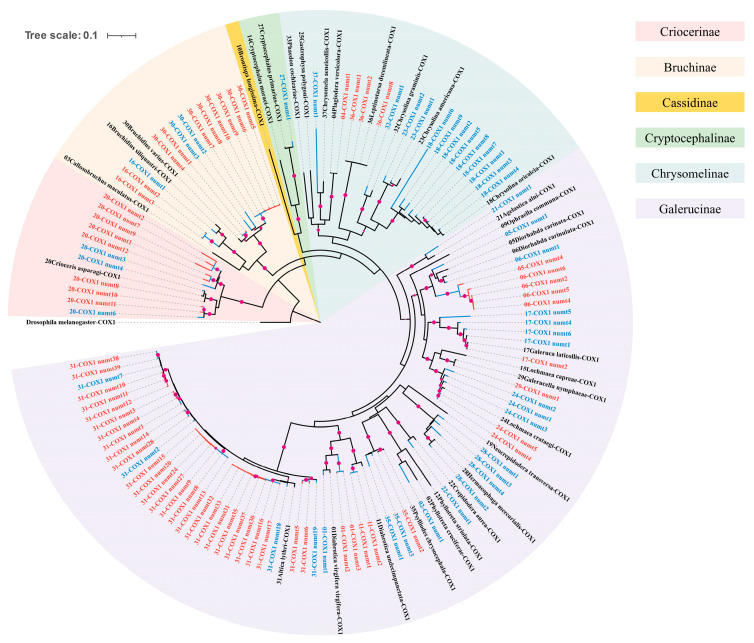
ML tree based on the *COX1* and NUMT (*COX1*+NUMT) dataset. The red branches represent the evolutionary history of nuclear genes, the blue branches reflect the mixed evolutionary history of nuclear and mitochondrial genomes, and the black branches correspond to the evolutionary history of the mitochondrial *COX1* gene. The colors of the tips correspond to the colors of the terminal branches. Some branches may appear visually distorted due to the short branch lengths. For clarity, bootstrap values greater than 75 are shown in this tree, with magenta-colored dots representing bootstrap values. The larger the dot, the higher the bootstrap value. The ML tree with specific bootstrap values and without branch lengths is presented in Appendix A.

**Table 1 insects-16-00150-t001:** Taxonomy and GenBank numbers of Chrysomelidae species.

Species Number	Species	Subfamily	Nuclear Genome	Mitochondrial Genome
1	*Diabrotica virgifera virgifera*	Galerucinae	GCF_917563875.1	KF658070.1
2	*Phyllotreta cruciferae*	Galerucinae	GCA_917563865.1	KX943506.1
3	*Callosobruchus maculatus*	Bruchinae	GCA_040182625.1	NC_053358.1
4	*Plagiodera versicolora*	Chrysomelinae	GCA_037013635.1	OR398224.1
5	*Diorhabda carinata*	Galerucinae	GCA_029229535.1	NC_042945.1
6	*Diorhabda carinulata*	Galerucinae	GCF_026250575.1	NC_042946.1
7 *	*Diorhabda sublineata* *	Galerucinae	GCF_026230105.1	/
8 *	*Diorhabda elongata* *	Galerucinae	GCA_026230145.1	/
9	*Ophraella communa*	Galerucinae	GCA_035357415.1	NC_039710.1
10	*Brontispa longissima*	Cassidinae	GCA_040580785.1	NC_053935.1
11	*Diabrotica undecimpunctata*	Galerucinae	GCA_040954645.1	CM082575.1
12	*Phyllotreta striolata*	Galerucinae	GCA_918026865.1	NC_045901.1
13 *	*Octodonta nipae* *	Cassidinae	GCA_034190945.1	/
14	*Cryptocephalus moraei*	Cryptocephalinae	GCA_946251905.1	OX276467.1
15	*Lochmaea capreae*	Galerucinae	GCA_949126875.1	OX421412.1
16	*Bruchidius siliquastri*	Bruchinae	GCA_949316355.1	OX438540.1
17	*Galeruca laticollis*	Galerucinae	GCA_963921935.1	OY998187.1
18	*Chrysolina oricalcia*	Chrysomelinae	GCA_944452925.2	OX101830.2
19	*Neocrepidodera transversa*	Galerucinae	GCA_963243735.1	OY725357.1
20	*Crioceris asparagi*	Criocerinae	GCA_958507055.1	OY293835.1
21	*Agelastica alni*	Galerucinae	GCA_950111635.2	OX467720.1
22	*Crepidodera aurea*	Galerucinae	GCA_949320105.2	OX439482.1
23	*Chrysolina americana*	Chrysomelinae	GCA_958502065.1	OY293422.1
24	*Lochmaea crataegi*	Galerucinae	GCA_947563755.1	OX387438.1
25	*Gastrophysa polygoni*	Chrysomelinae	GCA_963576655.1	OY755094.1
26 *	*Chrysolina haemoptera* *	Chrysomelinae	GCA_958298965.1	OY282604.1
27	*Cryptocephalus primarius*	Cryptocephalinae	GCA_963576515.1	OY754957.1
28	*Hermaeophaga mercurialis*	Galerucinae	GCA_951812935.1	OX638367.1
29	*Galerucella nymphaeae*	Galerucinae	GCA_963978555.1	OZ021720.1
30	*Bruchidius varius*	Bruchinae	GCA_964204745.1	OZ123453.1
31	*Altica lythri*	Galerucinae	GCA_964028335.1	OZ034812.1
32	*Chrysolina graminis*	Chrysomelinae	GCA_964197785.1	OZ078258.1
33	*Phaedon cochleariae*	Chrysomelinae	GCA_918026855.4	OU815794.1
34 *	*Diabrotica balteata* *	Galerucinae	GCA_918026665.1	OU815636.1
35	*Psylliodes chrysocephala*	Galerucinae	GCA_927349885.1	KX943483.1
36	*Leptinotarsa decemlineata*	Chrysomelinae	GCA_024712935.1	MZ189364.1
37	*Chrysomela aeneicollis*	Chrysomelinae	GCA_027562985.1	OP787486.1

* The species was not included in the subsequent NUMT analysis.

## Data Availability

Data are derived from public domain resources.

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
