# Peer review of "A Genome-Wide Analysis of Nuclear Mitochondrial DNA Sequences (NUMTs) in Chrysomelidae Species (Coleoptera)"

_insects, 2025, doi:10.3390/insects16020150_

Round 1

Reviewer 1 Report

Comments and Suggestions for Authors

A well written and interesting manuscript. NUMTs are rarely considered when carrying out species identification and phylogenetics and this paper highlights this potential issue. The introduction sets the scene well outlining the issues. The methodology is thorough with care taken to avoid biases in the data sets used and utilising both amino acid and nucleotide sequences add to the robustness of the analysis and allows a deeper understanding of the evolutionary processes. The picture drawn is highly complex but the authors have managed to draw out key point/conclusions from the data. I have just a couple of suggestions:

lines 154-162 describes the 5 NUMT categories, C2 for significantly large length variations and C5 for small length variation -  can you define what determines these two (include specific thresholds).

ATP8 was first described as not being a suitable phylogenetic marker due to severe substitution saturation yet in the conclusion is listed as a good marker (based on low levels of NUMTS) - could this contradiction be reconciled/clarified

COX1 appears to have a large number of NUMTs yet you suggested this has minimal interference - as this is probably the most commonly used locus for phylogenetic analysis I think the rationale for its use as a reliable and stable marker needs clarification.

Reviewer 2 Report

Comments and Suggestions for Authors

Overview and general recommendation:

This is an interesting work on the Analysis of Nuclear Mitochondrial DNA Sequences of Chrysomelidae Species, and the manuscript is well-written to some degree. If the authors modified the following comments carefully, I think it can be accepted and published.

Major comments:

The study mentioned phylogenetic analysis of Chrysomelidae Species and constructed a Maximum Likelihood tree, but did not construct a Bayesian tree. I suggest that the authors add a Bayesian tree for comparison, in order to gain a more comprehensive understanding of the evolutionary relationships between species or genes.

Minor comments:

1). Page 2, line 45-48: While the features of mtDNA are well-stated, the phrase "rapid evolution" may oversimplify mtDNA evolution. The evolutionary rate varies among different regions of mtDNA, with protein-coding regions evolving more slowly than the control region.

2). Page 2, line 62-64: The phrase "introduced complexities" could be more specific about how NUMTs complicate these research areas. It would be clearer to briefly mention the challenges they cause (e.g., misidentification, interference with sequence alignment, or erroneous phylogenetic trees).

3). Page 3, line 106-108: The phrase "individual or partial mitochondrial gene fragments" is somewhat vague. It could be more specific about which genes are typically studied in such research, which would provide better context for the reader.

4). Page 3, line 121: While RNA-seq is a valuable source for genomic data, the specific type of RNA-seq (e.g., transcriptome, whole-genome RNA-seq) could be clarified, as different RNA-seq approaches can lead to varying levels of completeness and accuracy for mitochondrial genome assemblies.

5). Page 5, line 177-179: The phrase "a suite of tree-building tools" is used in the context of PhyloSuite, but it would be clearer if you briefly mentioned what types of tools these are, or at least gave an indication of what tools are used within PhyloSuite.

6). Page 5, line 180: TrimAI is mentioned as a tool to improve sequence quality, but explaining why trimming is necessary (e.g., to remove poor-quality or ambiguous sequence regions) would provide more context.

7). Page 5, line 199-104: The explanation of the Bensasson method could be more detailed regarding how the chi-square test is used. It would be helpful to describe what the chi-square test specifically tests for and why this test is critical for determining whether the nucleotide substitutions are neutral.

8). Page 6, line 216-218: The input for CODEML is mentioned, but it could be clarified that both the aligned sequences and the phylogenetic tree topology are necessary for the calculation of dN/dS, and why the phylogenetic tree topology is required.

9). Page 6, line 223-224: The subject "ND1" is singular, so "use" should be changed to "uses."

10). Page 6, line 225-226: The phrase "is unusual" might be improved by providing more context or emphasizing its significance in the broader analysis.

11). Page 6, line 236: The semicolon at the end (";") is unnecessary and incorrectly placed.

12). Page 6, line 243-244: The term "significant differences" could be vague if not supported by specific statistical methods or values.

13). Page 6, line 245: The phrase "as many as 803 NUMTs" implies a subjective tone; consider being more neutral.

14). Page 6, line 250-251: It might be more precise to specify the types of chromosomal locations (e.g., intergenic regions, exons).

15). Page 8, line 273-301: The phrase "Figure S1: Dynamics of NUMTs generation and evolution along mitochondrial (mt) lineages (left) and the resulting phylogenetic reconstructions (right);" is repeated multiple times throughout the text. This repetition makes the writing less concise and could confuse readers.

16). Page 9, line 308-309: The sentence "ATP8-NUMTs were only reduced by 13.56%, indicating relatively stable lengths" is somewhat unclear. It would be more informative if you specify why ATP8-NUMTs are more stable compared to other NUMTs.

17). Page 9, line 329-331: The sentence "the significant difference in C1-NUMTs abundance between ATP8 and other PCGs across these species was highlighted, as determined by the Wilcoxon signed-rank test" could be more explicit. It's important to mention whether the comparison was conducted pairwise or across multiple PCGs.

18). Page 10, line 365-366: The term "inconsistencies" is somewhat vague. Its not clear whether the inconsistencies refer to differences in tree topology or some other aspect of the phylogenetic analysis. Clarifying this point would improve the precision of the description.

19). Page 11, line 386-387: The phrase "NUMTs were compared with their corresponding PCGs" lacks clarity in terms of how the comparison was made. It is important to specify whether this comparison was done at the sequence level, codon level, or some other measure.

20). Page 12, line 432-434: The statement on the diversity of gene lengths (ND5 gene example) could benefit from more specific details. For example, is the variation in gene length (e.g., the difference between 1668 bp and 1722 bp in ND5) considered biologically significant? The text doesn't elaborate on why this variation exists or its implications for gene function or phylogenetic relationships.

21). Page 13, line 480: The text provides data on the p-distance of the ATP8 gene between species but does not offer any explanation of why a p-distance greater than 3% is considered significant or what it implies in terms of evolutionary relationships between species. More context is needed to justify the p-distance threshold.

22). Page 14, line 545-547: The phrase "Differences between amino acid-based and nucleotide-based trees suggest that sequence type and saturation levels may impact phylogenetic signals, leading to biases in both topology and support values" is a bit general. The text should clarify how amino acid and nucleotide trees differ in their impact on phylogenetic analysis and provide specific examples or explanations.

23). Page 15, line 579-580: The text states that the COX1+NUMTs dataset yields "more divergent phylogenetic relationships" but does not elaborate on why this happens or what "divergent" means in this context. A more specific explanation of the term "divergent" in this context would be helpful.
